# The Impact of Sleep Quality on Mood Status and Quality of Life in Patients with Alopecia Areata: A Comparative Study

**DOI:** 10.3390/ijerph192013126

**Published:** 2022-10-12

**Authors:** Manuel Sánchez-Díaz, Pablo Díaz-Calvillo, Alberto Soto-Moreno, Alejandro Molina-Leyva, Salvador Arias-Santiago

**Affiliations:** 1Dermatology Unit, Hospital Universitario Virgen de las Nieves, 18002 Granada, Spain; 2Biosanitary Institute of Granada (ibs.GRANADA), 18002 Granada, Spain; 3Trichology Clinic, Hospital Universitario Virgen de las Nieves, 18002 Granada, Spain; 4Dermatology Department, School of Medicine, University of Granada, 18016 Granada, Spain

**Keywords:** Alopecia Areata, sleep quality, quality of life, mood status disturbances

## Abstract

Alopecia Areata (AA) is a chronic condition which has been associated with poor quality of life and mood status disturbances. The aim of this study is to compare the sleep quality between AA patients and controls, and to analyze the impact of poor sleep quality on patients with AA regarding mood status disturbances, quality of life and sexuality. A cross-sectional study including patients suffering from mild-to-severe AA and sex- and age-matched healthy controls was performed. Socio-demographic and clinical variables, sleep quality, quality of life, sexual disfunction, anxiety, depression and personality were collected using validated questionnaires. A total of 120 participants (60 patients and 60 controls) were included. Patients with AA showed worse sleep scores than controls (*p* = 0.003), as well as higher rates of anxiety and depression (*p* < 0.05). After a multivariate analysis, a worse sleep quality was found to be linked to anxiety, depression, a poorer quality of life and a type D personality score independently of the disease severity. In light of the results, patients with AA have a worse sleep quality than healthy controls. A poor sleep quality is associated with anxiety, depression and a worse quality of life, therefore being a general marker of a poor quality of life. Screening for sleep disturbances in specialized units could be useful to detect patients who could benefit from additional psychological support.

## 1. Introduction

Alopecia Areata (AA) is an autoimmune, common and non-scarring type of alopecia with an unpredictable evolution [1]. The severity ranges from mild cases with well-defined alopecic patches in the scalp to severe cases of total loss of all hair body [2]. Given the potential impact that hair loss can have on self-image and self-esteem, a poorer quality of life has been described in AA patients [3], as well as increased rates of anxiety and depression [4].

On the other hand, insomnia and sleep disorders are frequent among the general population [5] and have been associated with a variety of cardiovascular [6], neurologic [7] or psychiatric [8] complications. Moreover, a worse sleep quality has been described in a variety of chronic skin diseases such as psoriasis [9], atopic dermatitis [10] or chronic spontaneous urticaria [11]. In this regard, recent studies [12] identify a complex interplay among various physiological, psychosocial and behavioral factors mediating the relationship between sleep loss and chronic inflammatory skin diseases. Moreover, some studies have shown an increased risk of developing AA in patients with sleep disorders [13], while others do not detect this association [14].

Moreover, a worse sleep quality has also been associated with a poor quality of life and with higher rates of anxiety and depression in different skin diseases, including psoriasis [15], atopic dermatitis [10] or chronic spontaneous urticaria [16].

Since the potential association of a worse sleep quality with a poor quality of life, mood status disturbances and personality has not been studied to date in patients with AA, the aims of the present study are: (a) to compare the prevalence of sleep quality impairment in patients with AA and healthy controls; (b) to analyze the relationship between sleep quality impairment and mood status disturbances in patients suffering from AA; and (c) to evaluate the potential impact of sleep quality impairment on different aspects of the quality of life and personality traits.

## 2. Materials and Methods

Design: Cross-sectional study including patients suffering from AA, from mild to severe cases, and sex- and age-matched healthy controls.

Patients: Patients included in the study were recruited from two sources: (a) patients who received health care in the Trichology Clinic of the Hospital Universitario Virgen de las Nieves, who were offered to complete an online questionnaire after their protocolized follow-up consultation; and (b) patients who were contacted by e-mail by “Asociación de Alopecia de Madrid”, the official Spanish patient association for patients with AA. These patients were offered to complete the online version of the questionnaire. Patients were recruited between January 2021 and June 2022. The questionnaire for the two sources of patients was identical and was completed using an online questionnaire.

Controls: A healthy volunteer without dermatological disease was matched for sex and age (±5 years) to each selected AA case and was recruited among the companions of dermatological patients. The chosen controls had to have no diagnosed acute or chronic diseases, including even asymptomatic ones.

Inclusion criteria: The inclusion criteria for patients were as follows: (a) patients with a clinical diagnosis of AA, of all degrees of severity and with any type of treatment; (b) patients aged 18 years old or older; and (c) informed consent to be included in the study. In the case of controls, inclusion criteria consisted of: (a) healthy volunteers recruited among the companions of patients seen in the dermatology department; (b) people aged 18 years old or older; and (c) informed consent to be included in the study.

Exclusion criteria: The exclusion criteria were: (a) refusal from the patient or control to participate in the study and (b) patients or controls who suffered from any other major disease which could affect their quality of life. Diseases considered were: (i) active oncological diseases; (ii) any cardiopulmonary, neurologic, digestive, metabolic, musculoskeletal or urinary disease limiting daily activity or generating a significant symptomatology; (iii) psychiatric disorders existing prior to the onset of urticaria; and (iv) skin diseases other than urticaria which cause significant quality of life impairment.

Ethics: The present study was approved by the Research Ethics Committee of “Hospital Universitario Virgen de las Nieves” (internal code 1859-N-20) and is in accordance with the Declaration of Helsinki.

### 2.1. Variables of Interest

#### 2.1.1. Main Variables

The main variables included variables related to the severity of the disease and those related to the quality-of-life assessment:Variables related to the severity and characteristics of the disease: Severity of Alopecia Tool II (SALT II): it was used as an objective measure of the severity of the disease. It represents the percentage of the scalp affected by AA [17].The presence of AA totalis (affecting the whole surface of the scalp) or AA universalis (affecting all of the hair of the body surface).The age of onset, disease duration, date of diagnosis and current treatments were collected.Variables related to quality of life, anxiety and depression, sexual dysfunction and type D personality (TDp). The following validated questionnaires were collected:Pittsburgh Sleep Quality Index (PSQI) Questionnaire: This is a validated questionnaire to study the quality of sleep of patients. It consists of different questions in which the patient must mark one of the multiple answers offered. The global score is scored from 0–21 points, with 21 being the number that implies the greatest impairment of sleep quality. A global score greater than 5 is considered relevant from the point of view of sleep quality impairment [18].Dermatology Life Quality Index (DLQI): It is an indicator of the general dermatologic quality of life in patients over 16 years of age. The questionnaire consists of 10 questions that are scored on a Likert scale from 0 to 3 each, with 0 being the least affected and 30 the most affected. The questions refer to the last 7 days [19].Hospital Anxiety and Depression Scale (HADS): This validated questionnaire is composed of 14 statements in which the patient must show the degree of agreement/disagreement, scoring each question using an adapted Likert scale. It is subdivided into two scales, with odd-numbered questions being scored for anxiety and even-numbered questions for depression. A score ≥8 on any of the subscales was considered indicative of anxiety or depression, respectively [20].DS14 Questionnaire: It was used to evaluate the presence of TDp. It consists of a Likert-type questionnaire composed of 14 items, 7 for negative affectivity and 7 for social inhibition. Each response is answered with values between 0 (completely false) and 4 (completely true). Scores for social inhibition, negative affectivity and the global DS14 score were calculated. Moreover, a score ≥10 in both spheres is established as a cut-off point as an indicator of TDp [21,22].International Index of Erectile Function (IIEF-5) [23] and Female sexual function Index (FSFI-6) [24] questionnaires: They were used to collect data on sexual dysfunction in men and women. The IIEF-5 covers all five spheres of sexual function in the male, and a score ≤21 was considered significant. The FSFI-6 assesses the six items of female sexual function, and a score ≤19 was established as indicative of dysfunction.

#### 2.1.2. Other Variables

Socio-demographic, biometric and clinical variables, including age, sex, comorbidities, previous treatments for AA, marital status, and educational level, were recorded by questionnaires. Specific questions regarding all exclusion criteria were included, which were mandatory to answer.

### 2.2. Statistical Analysis

Descriptive statistics were used to evaluate the characteristics of the sample. The Shapiro-Wilk test was used to assess the normality of the variables. Continuous variables are expressed as a mean and standard deviation (SD). Qualitative variables are expressed as relative and absolute frequency distributions. The χ2 test or Fisher’s exact test were used, as appropriate, to compare nominal variables, and the Student’s t-test or Wilcoxon-Mann-Whitney test were used to compare nominal and continuous data. To explore possible associated factors, a simple linear regression was used for continuous variables. The β coefficient and SD were used to predict the log odds of the dependent variable. A multivariate analysis was performed to confirm the association found in the simple analysis. Statistical significance was considered if *p* values were less than 0.05. Statistical analyses were performed using JMP version 14.1.0 (SAS institute, North Carolina, USA).

## 3. Results

### 3.1. Sociodemographic and Clinical Features of the Sample

A total of 120 participants were included in the study: sixty patients suffering from AA and sixty sex- and age-matched healthy controls. Most of the patients were recruited from the Trichology Clinic (66.7%, 40/60). No differences were found between either recruiting source in terms of sex, age, marital status, educational level or severity of the disease (*p* > 0.50). When comparing patients and controls, no differences were found in terms of sex, age, marital status, educational level or occupation (*p* > 0.30) (Table 1 and Table 2).

The mean age was around 39 years in both patients and controls, with a majority of female participants (75%, 45/60). Regarding the clinical characteristics of the patients suffering from AA, the mean basal SALT score was 56.74% (SD 42.01), with 36.7% (22/60) of the patients suffering from AA universalis. The mean disease duration was 9.83 years (SD 0.49). Finally, patients receiving most treatment modalities for alopecia areata were included, inclusive of a subgroup of patients with no current treatment (Table 1 and Table 2).

### 3.2. Sleep Quality, Mood Status and Type D Personality in Patients and Controls

Univariate analyses were performed to compare sleep quality, mood status, and components of the type D personality between patients and controls.

Regarding sleep quality, patients with AA showed higher PSQI scores than controls (*p* = 0.003), indicating a worse quality of sleep. Moreover, when classified as “poor sleep quality” or “normal sleep quality”, the proportion of patients with a poor sleep quality was significantly higher among patients with AA than among controls (86.7% vs. 70%, *p* = 0.02). In fact, having alopecia areata increased the likelihood of poor sleep quality by 1.23-fold (prevalence ratio 1.23, Confidence Interval 95% 1.02–1.50).

Regarding mood status disturbances, it was shown that mood disturbances were significantly higher in patients with AA when compared to controls: HADS–Anxiety and HADS–Depression scores were higher in AA patients (*p* < 0.001) (Table 1). Regarding personality traits, both components of type D personality, social inhibition and negative affectivity, were found to score higher in patients than in controls, as did the global DS14 score (*p* < 0.001). Moreover, the prevalence of type D personality was significantly higher in patients with AA (Table 1). The rest of the comparisons between patients and controls can be seen in Table 1.

### 3.3. Sleep Quality Impairment in Patients with AA, Associated Clinical Factors

Univariate analyses were performed to explore the potential impact of greater PSQI scores, as a measure of a worse quality of sleep, in patients with AA (Table 3). No association was found between a worse quality of sleep and sex, age, disease duration, marital status or educational level (*p* > 0.30). Moreover, neither the severity of the disease, measured as a SALT score, nor alopecia universalis were associated with PSQI (*p* > 0.20).

On the other hand, a worse quality of life score (DLQI) correlated with a worse quality of sleep (beta = 0.28, *p* < 0.001), as did the components of type D personality: the social inhibition score (beta = 0.23, *p* = 0.001) and negative affectivity score (beta = 0.24, *p* < 0.001). This correlation was also found for the DS14 score and type D personality (*p* < 0.001). Finally, both anxiety and depression scores were related to a worse quality of sleep in patients with AA (*p* < 0.05).

A multivariate analysis was performed to confirm the result of the univariate analysis. It was found that a worse quality of life was associated with worse anxiety scores (beta = 0.57, *p* = 0.001). Moreover, worse depression scores, a worse quality of life (DLQI) and a type D personality score (DS14) were found to be near statistical significance (*p* = 0.07, *p* = 0.05 and *p* = 0.09, respectively). These associations were independent from the sex and severity of the disease (*p* > 0.15).

## 4. Discussion

A worse sleep quality has been described in a variety of skin diseases, including psoriasis [15], atopic dermatitis [10] or chronic spontaneous urticaria [11]. However, to date, the potential association and implications of a worse sleep quality in patients with AA have not been properly described.

This study first aimed to compare the prevalence of poor sleep quality in a sample of patients with AA and healthy controls. We have found that poor sleep quality scores seem to be higher in patients than in controls. In this regard, there are few studies addressing the issue of sleep quality in patients suffering from AA. One study exploring sleep quality in AA [25] was in line with the results of the present report: patients with AA seem to have higher PSQI scores than controls, as well as higher rates of daytime sleepiness. Moreover, as the present study shows, anxiety and depression are also more frequent among patients with AA than among controls. Therefore, it could be hypothesized that AA, as a chronic disorder, could lead to higher rates of anxiety and depression, which could be responsible for the worse quality of sleep in AA patients, or, on the contrary, that poor sleep quality could be responsible for higher rates of anxiety and depression in patients with AA.

Moreover, some studies have addressed the issue of the biological link between sleep disturbances and the development of AA. For instance, there is evidence that circadian clock genes, such as *Clock* or *Bmal1* genes, are highly expressed in hair germ progenitors in early anagen. It has been shown that the deficiency of these genes would cause a delay in anagen progression, which might be related to the aggravation of alopecia areata due to sleep disturbances [26]. On the other hand, it has been proven that biological clock disturbances are capable of altering the immune response, which may facilitate the development of pathologies such as alopecia areata [27].

On the other hand, the present study also aimed to assess the impact of sleep quality on the impairment of the quality of life and mood status in patients with AA. In light of the results, sleep quality should be considered as an indicator of a poorer quality of life and mood status disturbances in patients with AA, as it correlates with poor quality of life, anxiety and depression scores independently of the severity of the disease. Similar studies [25] have found consistent results: sleep quality scores do not seem to correlate with the severity of the disease, whereas they do with anxiety and depression rates. It is possible that the remitting and recurrent course of AA results in a poorer sleep quality as well as in higher rates of anxiety and depression, related to the chronic condition itself, without being directly related to the severity of the disease at a specific point in time. A longitudinal assessment of patients would be of interest to properly correlate the severity of the disease throughout the patients’ life history with the occurrence of sleep quality disturbances.

Similar studies have been performed regarding sleep quality and other skin conditions, such as psoriasis [9,15]. Interestingly, in these cases, a significant correlation was found between the severity and symptoms of the disease (pain, tender points), as well as subjective wellness (visual analogue scale) and sleep quality indicators. Given that AA is a chronic disease that usually lacks a relevant symptomatology, the relationship between the disease severity and sleep quality deficit may be weaker or non-existent.

The results of the present study should be taken into consideration under the limitations which it may have: (1) the cross-sectional design, which makes it impossible to assess causality; (2) the limited sample size, which could have limited the finding of statistical significance; (3) the selection of the patients from a specialized unit, which could lead to an underrepresentation of patients with milder forms who do not reach this level of medical care.

## 5. Conclusions

Since poor sleep quality can be more frequent among patients with AA when compared to controls, specific screening questionnaires would be of interest in specialized units. Moreover, since it seems to be associated with a poorer quality of life and higher rates of anxiety and depression independently of the disease severity, it could serve as a general marker for the holistic evaluation of patients with AA.

## Figures and Tables

**Table 1 ijerph-19-13126-t001:** Clinical characteristics of the patients suffering from AA.

Clinical Characteristics			
AA Patients (*n* = 60)			
Basal SALT score (%)	56.74 (SD 42.01)	Alopecia Areata Universalis (%)	36.7% (22/60)
Evolution time of the disease (years)	9.83 (SD 0.49)	Dermatology Quality of Life Index (DLQI)	6.89 (SD 7.07)
Current treatments for AA	No treatment: 25% (15/60)	Topical treatments: 36.7% (22/60)	Intralesional corticosteroids: 5% (3/60)
	Oral corticosteroids: 16.7% (10/60)	Immunosuppresive agents: 10% (6/60)	Janus Kinase inhibitors: 6.7% (4/60)

**Table 2 ijerph-19-13126-t002:** Comparative socio-demographic characteristics, mood disturbances, sexuality and personality measurements of the participants.

**Socio-Demographic Characteristics**			
	**Alopecia Areata** **(*n* = 60)**	**Controls** **(*n* = 60)**	***p* Value**
Age (years)	39.68 (SD 13.15)	40.21 (SD 12.79)	0.82
Sex (female)	75% (45/60)	75% (45/60)	1.0
Marital Status (couple)	63.3% (38/60)	73.3% (44/60)	0.45
Educational level (superior)	66.6% (40/60)	58.3% (35/60)	0.35
Occupation (Employed)	61.67% (37/60)	61.67% (37/60)	1.0
**Sleep Quality Impairment, Mood Status Disturbances and Personality Traits**			
	**Alopecia Areata ** (***n* = 60)**	**Controls** **(*n* = 60)**	***p* Value**
Pittsburg Sleep Quality Index (PSQI)	10.60 (SD 0.59)	8.10 (SD 0.59)	0.003
Sleep Quality Impairment (PSQI > 5)	86.67% (52/60)	70% (42/60)	0.02
HADS–Anxiety score	8.45 (SD 0.52)	4.13 (SD 0.52)	<0.001 *
Anxiety (%)	55% (33/60)	15% (9/60)	<0.001 *
HADS–Depression score	13.80 (SD 0.48)	2.15 (SD 0.48)	<0.001 *
Depression (%)	96.7% (58/60)	6.7% (4/60)	<0.001 *
IIEF score (male)	21 (SD 1.01)	21.27 (SD 0.92)	0.84
Sexual dysfunction (male)	33.3% (5/15)	40% (6/15)	0.33
FSFI score (female)	13.95 (SD 1.42)	16.35 (SD 1.47)	0.24
Sexual dysfunction (female)	66.7% (30/45)	42.2% (19/45)	0.
Social inhibition score	8.63 (SD 0.93)	6.2 (SD 0.93)	0.06
Negative affectivity score	15.73 (SD 1.01)	8.73 (SD 1.01)	<0.001 *
DS14 global score	24.36 (SD 1.66)	14.93 (SD 1.66)	<0.001 *
Type D personality (%)	35% (21/60)	15% (9/60)	0.01 *

DLQI; Dermatology Life Quality Index; DS14: Scale for measuring type D personality; FSFI: Female Sexual Function Index; HADS: Hospital Anxiety and Depression Scale; IIEF: International Index of Erectile Function; SALT: Severity of Alopecia Tool; SE: Standard Error. * Results which were within statistical significance limits.

**Table 3 ijerph-19-13126-t003:** Univariate analysis for PSQI index. Associated clinical and socio-demographic factors.

Factors	Simple Analysis		Multivariate Analysis	
	Mean/Beta	*p* Value	Mean/Beta	*p* Value
Sex	Male: 9.60 (SE 1.16)	0.32	0.88 (SE 0.60) (Male)	0.16
	Female: 10.93 (SE 0.67)			
Age	−0.03 (SE 0.04)	0.50	-	-
Duration of disease	0.008 (SE 0.01)	0.87	-	-
Marital Status	Single: 10.60 (SE 0.53)	0.99	-	-
	Couple: 10.59 (SE0.97)			
Educational level	Superior studies: 10.80 (SE 0.71)	0.63		
	Basic or no studies: 10.20 (SE 1.01)			
SALT score	0.01 (SE 0.01)	0.25	0.01 (SE 0.01)	0.37
Alopecia universalis	Yes: 11.5 (SE 0.97)	0.24		
	No: 10.02 (SE 0.76)			
DLQI score	0.28 (SE 0.06)	<0.001 *	0.15 (SE 0.07)	0.05 *
DS14 score	0.16 (SE 0.03)	<0.001 *	0.09 (SE 0.05)	0.09
Social inhibition score	0.23 (SE 0.07)	0.001 *	-	-
Negative affectivity score	0.24 (SE 0.06)	<0.001 *	-	-
Type D personality	TDp+: 12.6 (SE 0.93)	<0.001 *	-	-
	TDp−: 9.51 (SE 0.58)			
HADS–Anxiety score	0.59 (SE 0.10)	<0.001 *	0.57 (SE 0.16)	0.001 *
HADS–Depression score	0.41 (SE 0.12)	0.001 *	0.34 (SE 0.18)	0.07
IIEF score (male patients)	−0.31 (SE 0.24)	0.23	-	-
FSFI score (female patients)	−0.03 (SE 0.08)	0.71	-	-
R^2^			0.47	

DLQI: Dermatology Life Quality Index; DS14: Scale for measuring type D personality; FSFI: Female Sexual Function Index; HADS: Hospital Anxiety and Depression Scale; IIEF: International Index of Erectile Function; SALT: Severity of Alopecia Tool; SE: Standard Error. * Results which were within statistical significance limits.

## Data Availability

Data are available upon reasonable request.

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
