# Peer review of "The Impact of Sleep Quality on Mood Status and Quality of Life in Patients with Alopecia Areata: A Comparative Study"

_ijerph, 2022, doi:10.3390/ijerph192013126_

Round 1
Reviewer 1 Report
1.This topic in very interesting, but I think that authors dismissed existing data. Several studies investigated the risk of developing AA in patient with sleep disorders and found that sleep disorders are associated with increased risk of developing AA (Seo HM, et al Sleep 2018; Dai YX, et al 2020). Another study showed no relationship between sleep quality and risk of AA (Inui S, et al 2014). Several studies have investigated the mechanisms of sleep disturbance in AA. The influence of biologic clock dysfunction leading to Th1 response was suggested by Takata E, et al , 2013. There is evidence that circadian clock genes are expressed in hair germ progenitors in early anagen. Decrease in clock genes CLOCK and BMAI1expression delays anagen progression, and this may be related to aggravation of AA in sleep disturbances(Lim KK, et al 2009). Deficiency of clock genes was linked to Th17 immune response and leading to AA (Lim KK, 2009). I think these data need to be discussed, and references need to be cited.
2. SQ1 (line 44) and TDp (p2 line 62)- abbreviation used without explanation.
3. line 86- from among - need to remove one of these words
4. In exclusion criteria major diseases were mentioned- how authors decided what is major?
5. Line 86 - evolution from disease onset- is it time from diagnosis or disease duration? What do you mean by evolution?
Author Response
Dear Reviewer,
We would like to thank you for your comments, as they allow us to improve the scientific quality of our work. Here, you can see a point-by-point response to your comments:
1) These references has been added and discussed both in the introduction section and the discussion section.
2) These mistakes have been corrected.
3) This mistake have been corrected.
4) A more extensive explanation about this issue has been added.
5) This mistake has been corrected.
Reviewer 2 Report
INTRODUCTION
The introduction is well written and covers the main issues.
There is one mistake:
Line 44. The acronym SQI is not explained.
MATERIALS AND METHODS
Design: in this section the authors repeat the objective of the study, which was formulated at the end of the Introduction section.
Inclusion criteria: The authors point out that one of the inclusion criteria for control cases was to be healthy. “Healthy” is an unspecific word….What means healthy for authors? Normal cholesterol levels? free of any disease? Please, specify the mean of healthy and how the authors corroborated it.
Variables: The authors mention a lot of variables, but not all of them were used finally in the study. I suggest to remove those no were analysed in this paper.
Line 89: The acronym TDp is not explained
Statistical analysis: line 131. There is a change in the font size.
RESULTS
In general, in the Results section, the authors show data, but they do not specify the statistical test used. They refer to the analyses used in previous section (Material and Methods), but I find specially relevant to include before the data, the statistic test used. It would make this section more comprehensible.
Tables: It was interesting to point out the analysis which are statistically significant. Usually it is used *. Tables include acronym not explain….(table 1 JAK. Table 2 SE)
Table 1 are too long and mix a lot of information. I suggest to divide the table 1, in two. One which contains the socio-demographic information and clinical characteristics, and another one with the first analysis.
DISCUSSION
It is very short and only one new reference is introduced.
The limitations of the study have not been included
REFERENCES
Review this section. The more of the references do not have DOI.
Author Response
Dear Reviewer,
We would like to thank you as your comments allow us to improve the scientific quality of our work. All the suggested changes have been performed. Below, you can see a point-by-point response to your comments.
Introduction: The mistakes have been corrected.
Material and methods:
- The repetition has been removed.
- The inclusion criteria for controls has been specified.
- The variables have been revised.
- Typing errors have been corrected.
Results:
- Tables have been improved to meet the reviewer's criteria.
- Table 1 has been divided.
- In order to make this section more easily readable, we would like to keep the current format of the results. In case it is considered essential, please let us know and we will modify the format by including the statistical tests before each result displayed.
Discussion:
- The review has been expanded with new studies and data and a limitations paragraph has been included.
References: References have been modified.